# BenchCLAMP: A Benchmark for Evaluating Language Models on Syntactic and Semantic Parsing

**Subhro Roy**[1]    **Sam Thomson**[1]    **Tongfei Chen**[1]    **Richard Shin**[1]
**Adam Pauls**[2*]    **Jason Eisner**[1]    **Benjamin Van Durme**[1]

[1]Microsoft Semantic Machines    [2]Scaled Cognition

[1]{subhrroy,sathomso,tongfeichen,eush,jeisner,bevandur}@microsoft.com
[2]adpauls@scaledcognition.com

## Abstract

Recent work has shown that generation from a prompted or fine-tuned language model can perform well at semantic parsing when the output is constrained to be a valid semantic representation. We introduce **BenchCLAMP**, a **Bench**mark to evaluate **C**onstrained **LA**nguage **M**odel **P**arsing, that includes context-free grammars for seven semantic parsing datasets and two syntactic parsing datasets with varied output representations, as well as a constrained decoding interface to generate only valid outputs covered by these grammars. We provide low, medium, and high resource splits for each dataset, allowing accurate comparison of various language models under different data regimes. Our benchmark supports evaluation of language models using prompt-based learning as well as fine-tuning. We benchmark eight language models, including two GPT-3 variants available only through an API. Our experiments show that encoder-decoder pretrained language models can achieve similar performance or surpass state-of-the-art methods for syntactic and semantic parsing when the model output is constrained to be valid.

## 1 Introduction

Large pretrained language models can achieve state-of-the-art performance on a host of NLP tasks when fine-tuned on target data [22, 28, 46, 13]. Models like GPT-3 [5], Codex [9] and T0 [31] have also shown impressive zero- and few-shot performance when prompted only with task descriptions and examples. Research on large language models is typically validated by performance on downstream NLP tasks. Past work has evaluated new pretrained language models on classification, extraction, and generation, among others [22, 13, 20]. However, parsing tasks are generally not considered a testbed for such evaluation. The outputs of parsing tasks are structured objects such as parse trees or code. State-of-the-art systems thus involve task- or dataset-specific model architectures and target representation constraints. Evaluating language models on parsing tasks test capabilities not captured by commonly used evaluation tasks.

Recently, Shin et al. [36] and Scholak et al. [32] have shown that generation from a fine-tuned or few-shot prompted language model can perform competitively in semantic parsing tasks, when the output of the language model is constrained to produce valid meaning representations. However, it is still challenging to set up constrained generation for a new dataset and language model due to the variation in output formalisms and model-specific tokenization. In this paper, we introduce a new benchmark called BenchCLAMP (Benchmark for Constrained Language Model Parsing) that covers nine parsing datasets with seven different output formalisms. We release context-free grammars for each dataset and provide a toolkit to perform

37th Conference on Neural Information Processing Systems (NeurIPS 2023) Track on Datasets and Benchmarks.

---

\* Work done while at Microsoft Semantic Machines.

efficient constrained decoding to generate valid output representations. Our benchmark reduces the barrier for language model developers to evaluate on parsing. The benchmark is available at `https://github.com/microsoft/semantic_parsing_with_constrained_lm`.

We benchmark eight pretrained language models using BenchCLAMP. We find that fine-tuning encoder-decoder pretrained language models can come close to or surpass the performance of state-of-the-art methods on all parsing datasets. Constrained generation via a domain-specific grammar provides performance gains for most fine-tuned language models. The improvement is high in low-resource settings but the relative improvement reduces when more training data becomes available. We find constrained decoding to be essential for few-shot prompted models and for tasks with complex constraints like constituency and dependency parsing. In both these cases, we find language models struggle to generate valid representations without constrained decoding. In addition, we ablate different ways to encode context in the input, prompt structure, and retrieval methods for few-shot prompted language models. We present a comprehensive study establishing concrete techniques to reliably use language models for syntactic and semantic parsing tasks.

## 2 Related Work

**Language Models for Semantic Parsing:** Recent work has shown that one can generate an analysis of a natural language sentence, such as a semantic parse, by asking a large language model to continue a prompt that includes the sentence [9, 18, 33]. We refer to this as "language model parsing." To avoid ill-formed analyses, it is possible to *constrain* the generation so that the generated output satisfies hard task-specific constraints. Shin et al. [36] showed that constrained generation from few-shot prompted GPT-3 and fine-tuned BART models outperformed task-specific semantic parsing architectures in low-resource settings. Scholak et al. [32] were able to achieve state-of-the-art performance in SQL prediction by fine-tuning a T5-3B model [28] and using constrained decoding. Recent work on AMR parsing has also shown positive results with sequence-to-sequence training with pretrained language model parameters [6, 3]. As the above works used different evaluation settings, it is hard to see which techniques work best under different data regimes.

**Language Models for Syntactic Parsing:** Syntactic parsing tasks like constituency and dependency parsing requires the outputs to be well aligned with the input. All input tokens need to be covered by the output constituency or dependency parse. As a result, most solutions for syntactic parsing has involved custom decoders or inference algorithms [14, 49, 53, 21]. However there has been some work on generating linearized representations of the output [15, 47, 19, 12].

**NLP Benchmarks:** Multiple benchmarks have been introduced to track progress on specific NLP tasks and to encourage multi-task learning with diverse datasets. The GLUE [42], SuperGLUE [43], BIG-bench [37] and HELM [20] are widely used by language model developers. However, these benchmarks focus on classification, span extraction and generation, and do not include structured prediction tasks like parsing. More recently, the UnifiedSKG [48] benchmark has been introduced that converts a suite of tasks requiring structured knowledge into text-to-text format. In contrast to their work, we cover a wide range of parsing tasks covering AMR, SMCalFlow, constituency and dependency parsing. While they focus on unconstrained generation, we develop grammars and constrained decoding interface to support valid representation generation.

## 3 Benchmark Details

### 3.1 Data Setup

BenchCLAMP includes nine popular parsing datasets with a varied set of meaning representation formalisms (details in Table 1). We create several splits for each dataset. For datasets that do not release public test sets, like SMCalFlow, Spider, and CoSQL, we treat the development set as the test set, and sample $10\%$ of the training set and treat it as the development set for splits creation.

1. We create **three low-resource train splits** of 500 examples, each uniformly sampled from the training portion of the dataset. We create a single low-resource development set of 50 examples sampled from the development portion of the dataset. We report mean of these splits.

2. We similarly create **a medium-resource train split** of 5000 examples paired with a development set of 500 examples.

Table 1: List of datasets covered by BenchCLAMP, along with evaluation metric and an example representation. All representations are linearized into a sequence that can be produced by a language model. We use task and dataset specific metrics to evaluate performance.

| Dataset | Metric | Example Representation |
|---|---|---|
| SMCalFlow [1] TreeDST [10] | Lispress Match | `(Yield (Event.start (FindNumNextEvent (Event.subject_?  (?~= "meeting")) 1L)))` |
| MTOP [18] | Exact Match | `[IN:Get_Message [SL:Type_Content video] [SL:Sender Atlas]]` |
| Overnight [45] | Denotation Match | `(call listValue (call getProperty en.block.block1 (string color)))` |
| Spider [50] CoSQL [51] | Test suite Execution | `SELECT born_state FROM head GROUP BY born_state HAVING count(*) >= 3` |
| AMR 2.0 [4] | Smatch | `(e/establish-01 :ARG1 (m/model :mod (i/innovate-01 :ARG1 (i2/industry))))` |
| PTB 3 [23] (Constituency parsing) | EVALB | `(S (NP (NNP Ms.)  (NNP Haag)) (VP (VBZ plays) (NP (NNP Elianti))) (.  .))` |
| UD-EWT [52] (Dependency parsing) | LAS | `(1-Read, root, 0) (2-some, obj, 1) (3-of, case, 6) (4-the, det, 6) (5-following, amod, 6) (6-links, nmod, 2)` |

3. We consider **a high-resource split** with the entire training set of the dataset, paired with the medium-resource development set.

To make it feasible for researchers to evaluate large pretrained models on BenchCLAMP, we randomly sample a smaller test set for datasets with large released test sets. Specifically, we sample 2000 examples from the test sets of SMCalFlow, TreeDST and MTOP datasets and evaluate test performance on this smaller set. We use the full test set for all other datasets. We also release a smaller randomly-sampled 100-example test set for each dataset to evaluate models accessed through costly API calls like GPT-3 and Codex. Results on a 100-example test set will have wide error bars and should be used with caution. See the Appendix for a discussion on result variance. We allow for the evaluation on full test sets of the datasets to compare with state of the art results.

For datasets that include dialogue interactions, we ensure that all turns of a dialogue belong to the same split. The Overnight train set was already small (< 5k examples), so we do not have a separate medium split for it. We release data splits for all domains of Overnight and all MTOP languages, but for brevity we benchmark on a single Overnight domain (blocks) and a single MTOP language (English) in this paper.

**Linearizing Representations:** We use the dataset representations as is for the MTOP, PTB-3 and the SQL datasets. For AMR, we use the setup provided by [40, 41] to linearize the representations into sequences for training, and to convert output model predictions to AMRs for evaluation. For SMCalFlow and TreeDST, we use the Lispress format (LISP-like serialization for programs) of the data released by [27]. We linearize dependency parses into a sequence of dependency triples. For the example in Table 1, our representation has the following form:

(Read, root, root) (some, obj, Read) (of, case, links) (the, det, links) (following, amod, links) (links, nmod, some)

To convert such a representation back to a dependency parse, we find head token indices for each token based on string match with the predicted head token. In case there are multiple mentions of the head token, we select the one that is closest to the token being considered. We can correctly roundtrip $95.7\%$ of parses in the test set using this approach. For completion, we also release a loss less linearization of dependency parses which is similar to the one above but with the head token replaced by the corresponding token index in the input sequence.

## 3.2 Grammars

We release context-free grammars for all datasets to constrain generation to valid meaning representations. The grammar creation process is specific to each dataset.

1. For SMCalFlow and TreeDST, we use the Lispress-format datasets released by Platanios et al. [27]. We create a non-terminal corresponding to each type present in the training data. For each (sub-)expression with type $t$, we add a production rule with the non-terminal for $t$ generating the non-terminals of its component (sub-)expression types, or component terminal plan fragments.

2. For MTOP, we add a non-terminal corresponding to each intent and slot. Each intent non-terminal generates an expression comprising slot non-terminals. Similarly each slot non-terminal can generate an expression with nested intent non-terminals. Slot non-terminals can also generate terminal strings copied from the input utterance.

3. We use a publicly available SQL grammar [2] for Spider and CoSQL. For each example, we add schema-specific constraints to the grammar to generate consistent table and column names. This is similar to "parsing without guards" in Scholak et al. [32].

4. For constituency parsing, we define a non-terminal for an expression and a constituent label. We add production rules where the label non-terminal can produce any of the constituent labels seen in the training data. The expression non-terminal can either produce terminal tokens or generate a constituent label coupled with a expression non-terminal. This context free grammar covers constituent parse tree representation shown in Table 1. To additionally ensure that all tokens in the input utterance are covered by the generated parse tree, we additionally maintain a state in our parsing algorithm during decoding, allowing tokens to appear in the order seen in the utterance, and allowing the generation to end only when all input tokens have been generated.

5. For dependency parsing, we extract the set of dependency relations from the training set and define a non-terminal that can produce any of them. We then define a non-terminal that can generate a sequence of triples each comprising two tokens from the utterance and a dependency relation. Similar to our approach with constituency parsing, we maintain a parse state during decoding to ensure all tokens from the utterance are covered in order in the generated output.

For all data splits, we use the full training data to derive the grammar. We envision that in realistic scenarios, the grammar will be provided by a domain developer, and hence will have complete coverage of the domain (even when some plan fragments might not have appeared in the low-resource dataset). We also add results with grammars induced from low-resource splits in section 5.5.

## 4 Experimental Setup

### 4.1 Language Models Evaluated

We use BenchCLAMP to fine-tune and evaluate five language models with varying number of parameters: T5-base (220M), T5-large (770M), T5-3B (3B) [28], BART-large (406M) [17] and CodeT5-base (220M) [44]. The input to our model is the utterance concatenated with the string representation of the context (conversation context, database schema, etc.), and the output is the target parse. We evaluate three large language models: GPT-3 [5], Codex [9] and Llama-2 [39][1], using few-shot prompting on the 100 example test sets. Unless otherwise state, we use the OpenAI API `text-davinci-001` for GPT-3 and `code-davinci-001` for Codex. For each input (utterance concatenated with context) we select a set of 20 relevant examples from the training set using BM25 [30] or a sentence transformer [29] based similarity model. We create a prompt using these examples, following the template in Shin et al. [36] and limiting the total length of the prompt to be 1500 tokens. This leaves room in GPT-3's buffer to generate an output of up to 548 tokens.

We release data splits for all domains of Overnight and all languages in MTOP. But for brevity, we benchmark on a single domain of Overnight (blocks) and a single language from MTOP (English). All other datasets used in the benchmark are in English. We evaluate few-shot prompted GPT-3, Codex, and Llama-2 on three BenchCLAMP datasets. All other models are evaluated on the complete evaluation suite.

---

[1]We use `llama-2-7b` in our evaluation. Llama-2 is served using vLLM [16] to create the same OpenAI API as GPT-based models.

We use the code released by Shin et al. [36] to support incremental constrained generation of semantic representations. This code maintains a chart according to Earley's algorithm [11] that can be used to determine the set of legal next tokens and can also be efficiently updated after a particular token is selected. We extend their method to support all autoregressive language models and sequence-to-sequence models. Unless otherwise mentioned, we always use constrained decoding.

Table 2: Performance of language models on SMCalFlow, TreeDST and MTOP. $^\dagger$ indicates few-shot prompted and evaluated on the 100 example test set; numbers above the horizontal bar are thus not comparable to those below. $^1$ Prompt exemplars retrieved by BM25; $^2$ by dense vector retrieval using SentenceT5-xxl. Remaining LMs are finetuned and evaluated on BenchCLAMP test sets. We report results with both constrained and unconstrained decoding to illustrate the contribution of constraints. Metrics are dataset-specific (see Table 1). The best score in each column is boldfaced.

| LM | Grammar Constraints | SMCalflow | | | TreeDST | | | MTOP (en) | | |
|---|---|---|---|---|---|---|---|---|---|---|
| | | Low | Med | High | Low | Med | High | Low | Med | High |
| GPT-3$^{\dagger 1}$ | Yes | 26.0 | 48.0 | 49.0 | 35.7 | 54.0 | 53.0 | 46.3 | 56.0 | 57.0 |
| Codex$^{\dagger 1}$ | Yes | 36.7 | 55.0 | 56.0 | 46.3 | 61.0 | 62.0 | 53.3 | 68.0 | 72.0 |
| Llama-2-7b$^{\dagger 1}$ | No | 25.0 | 35.0 | 44.0 | 31.0 | 50.0 | 59.0 | 43.0 | 60.0 | 62.0 |
| GPT-3$^{\dagger 2}$ | Yes | 33.0 | 45.0 | 52.0 | 38.7 | 56.0 | 59.0 | 50.0 | 62.0 | 64.0 |
| Codex$^{\dagger 2}$ | Yes | 39.3 | 52.0 | 62.0 | 47.7 | 58.0 | 64.0 | 55.7 | 67.0 | 70.0 |
| Llama-2-7b$^{\dagger 2}$ | No | 29.0 | 39.0 | 42.0 | 28.0 | 46.0 | 58.0 | 44.0 | 61.0 | 62.0 |
| T5-base | No | 38.2 | 67.5 | 77.6 | 57.2 | 84.4 | 89.3 | 54.7 | 79.3 | 84.5 |
| | Yes | 41.6 | 69.7 | 78.6 | 62.0 | 85.8 | 89.4 | 57.5 | 80.1 | 84.3 |
| CodeT5-base | No | 33.3 | 65.6 | 80.8 | 50.3 | 83.5 | 90.3 | 44.1 | 75.6 | 80.6 |
| | Yes | 37.3 | 67.5 | 81.1 | 56.8 | 84.4 | 90.0 | 47.1 | 75.8 | 81.1 |
| BART-large | No | 36.1 | 68.1 | 82.2 | 52.0 | 84.0 | 90.2 | 57.8 | 81.6 | 85.8 |
| | Yes | 42.5 | 71.4 | **83.0** | 61.1 | 86.4 | 89.8 | 61.7 | 82.1 | 85.3 |
| T5-large | No | 42.6 | 71.5 | 81.3 | 59.6 | 86.2 | 90.0 | 55.4 | 82.1 | 85.6 |
| | Yes | 46.3 | 73.1 | 82.1 | **64.2** | **87.2** | 90.1 | 59.3 | 82.5 | 85.3 |
| T5-3B | No | 43.5 | 73.8 | 82.6 | 58.5 | 85.9 | **90.7** | 60.9 | 83.2 | **86.3** |
| | Yes | **48.7** | **75.9** | **83.0** | 64.1 | **87.2** | 90.3 | **64.1** | **83.4** | 85.6 |

Table 3: Performance of fine-tuned language models on Spider, CoSQL and Overnight datasets. We report test suite execution accuracy [54] for the SQL datasets and denotation accuracy for Overnight.

| LM | Grammar Constraints? | Spider | | | CoSQL | | | Overnight (blocks) | |
|---|---|---|---|---|---|---|---|---|---|
| | | Low | Med | High | Low | Med | High | Low | High |
| T5-base | No | 30.8 | 54.1 | 55.6 | 24.0 | 39.9 | 40.8 | 63.9 | 63.7 |
| | Yes | 33.8 | 56.8 | 58.9 | 26.8 | 42.7 | 43.4 | **64.4** | 63.9 |
| CodeT5-base | No | 36.6 | 57.4 | 61.9 | 26.1 | 45.9 | 48.1 | 60.0 | 64.7 |
| | Yes | 37.6 | 58.0 | 62.2 | 27.3 | 46.2 | 48.2 | 60.4 | 65.2 |
| BART-large | No | 41.8 | 59.1 | 63.9 | 30.9 | 49.9 | 52.8 | 60.7 | 63.4 |
| | Yes | 42.5 | 62.7 | 63.9 | 29.1 | 48.8 | 51.5 | 61.2 | 63.7 |
| T5-large | No | 42.4 | 64.6 | 65.7 | 30.7 | 50.0 | 53.8 | 62.1 | 68.7 |
| | Yes | 44.1 | 65.5 | 66.5 | 32.1 | 52.4 | 55.5 | 62.8 | **68.9** |
| T5-3B | No | 46.4 | 68.4 | 70.9 | 32.6 | 54.7 | 53.4 | 62.8 | 66.2 |
| | Yes | **48.6** | **70.3** | **72.3** | **34.7** | **56.4** | **56.2** | 63.2 | 66.2 |

## 4.2 Format for Model Inputs

For experiments related to fine-tuned language models with SMCalflow and TreeDST with last user and agent utterance as context, the input to the model has the format $l \mid a \mid u$, where $u$ is the input natural language utterance, $l$ is the last user utterance, $a$ is the last agent utterance and $\mid$ is a separator

Table 4: Performance of fine-tuned language models on constituency (PTB-3), dependency (UD-EWT) and AMR paring. We report bracketing F1 using EVALB [34] for PTB-3, labeled attachment score (LAS) for UD-EWT, and Smatch [7] for AMR parsing.

| LM | Grammar Constraints? | PTB-3 | | | UD-EWT | | | AMR 2.0 | | |
|---|---|---|---|---|---|---|---|---|---|---|
| | | Low | Med | High | Low | Med | High | Low | Med | High |
| T5-base | No | - | - | - | - | - | - | 52.7 | 72.0 | 75.0 |
| | Yes | 83.1 | 93.1 | 94.6 | 80.2 | 88.2 | 89.4 | 51.3 | 72.0 | 75.0 |
| CodeT5-base | No | - | - | - | - | - | - | 48.0 | 66.0 | 74.0 |
| | Yes | 70.9 | 86.7 | 92.1 | 73.2 | 84.0 | 85.8 | 46.7 | 66.0 | 74.0 |
| BART-large | No | - | - | - | - | - | - | 57.0 | 74.0 | 81.0 |
| | Yes | 78.0 | 93.9 | 95.7 | **84.1** | 89.7 | 90.6 | 55.0 | 75.0 | 81.0 |
| T5-large | No | - | - | - | - | - | - | 57.3 | 76.0 | 81.0 |
| | Yes | **84.5** | **94.4** | 95.7 | 80.6 | 90.1 | 91.0 | 57.0 | 76.0 | 81.0 |
| T5-3B | No | - | - | - | - | - | - | **60.0** | **77.0** | 82.0 |
| | Yes | 77.6 | 93.9 | **96.2** | 83.1 | **90.4** | **91.3** | 59.0 | 77.0 | **83.0** |

symbol. When using only last agent utterance as context, the input is $a \mid u$, and for using no context, the input to the model is simply $u$.

We use the the following format for Spider and CoSQL: $c\,,d\,,u$, where $c$ is any conversational context if applicable, $d$ is a rendering of the database schema with or without values and $u$ is the user utterance. We use the database schema representation used in Scholak et al. [32] for $d$. For $c$, we concatenate the past utterances in the conversational context with the separator symbol $|$.

Our few-shot prompting experiments use the prompt template of Shin et al. [36]. Given a language input, we retrieve relevant prompt examples and create a prompt with the following format:

```
Let's translate what a human user says into what a computer might say.

Human: {Prompt Example 1 Input}
Computer: {Prompt Example 1 Output}
Human: {Prompt Example 2 Input}
Computer: {Prompt Example 2 Output}
        ...
Human: {Current Input}
Computer:
```

### 4.3 Training Details

For fine-tuning experiments, we train the language models with batch size 32 for $10\,000$ steps using AdaFactor [35], saving a checkpoint every 5000 steps. We use 1000 linear warmup steps and then linear decay the learning rate to 0. We tune all models with learning rates $10^{-4}$ and $10^{-5}$, except for T5-3B for which we only used $10^{-4}$ to save compute. The best performing checkpoint on the dev set is used to report scores on the test set.

## 5 Results

### 5.1 Benchmarking Language Models

We show the performance of language models on BenchCLAMP datasets in tables 2, 3 and 4. We find that performance increases with model size for most fine-tuned language models. Few-shot prompting of GPT-3 and Codex are still not on par with fine-tuned models. For non-SQL datasets, even smaller language models reach close to the best performance in the high resource setting. However, for Spider and CoSQL, model size seems important in all data settings. This is likely because the model has to reason about the database schema to generate SQL queries, making it a harder learning problem. See sections 5.3 and 5.4 for details on how we use context in model inputs for these experiments. We skip

unconstrained generation results for constituency and dependency parsing since models often fail to generate valid parses for these tasks, and evaluating a valid substring is not supported by released evaluation tools. Entirely rejecting invalid parses leads to very low performance.

Table 5 compares our constrained T5-3B model with the best-performing models in the literature. We outperform state-of-the-art models on the SMCalFlow, TreeDST and Overnight-blocks datasets. For Spider and CoSQL, our scores are lower than the state-of-the-art despite using a similar constrained language model approach. This is likely because we use a general SQL grammar to constrain decoding, whereas the SQL in these datasets covers only a small fraction of the SQL grammar. We believe using a more constrained grammar will improve performance.The state-of-the-art method for AMR also uses a CLAMP model but with additional task specific pretraining. The best models for PTB-3 and UD-EWT are designed specific to the task. Our language model fine-tuning paired with constrained decoding achieves similar performance without any task specific modifications.

Table 5: Comparison of our fine-tuned T5-3B model with current state of the art models on full test sets. We report exact match accuracy for Spider and CoSQL to match the settings of previous work. The best score in each row is boldfaced.

| Dataset | Current State of the Art | Our T5-3B |
|---|---|---|
| SMCalFlow | 80.4 [27] | **83.7** |
| TreeDST | 88.1 [27] | **91.5** |
| MTOP (en) | **86.4** [26] | 85.7 |
| Overnight (blocks) | 65.2 [8] | **66.2** |
| Spider | **75.5** [32] | 72.2 |
| CoSQL | **56.9** [32] | 52.3 |
| AMR | **85.4** [3] | 83.0 |
| PTB 3 | **96.4** [38] | 96.2 |
| UD-EWT | **91.5** [24] | 91.3 |

## 5.2 Effect of Constraints

Decoding constrained by a grammar is essential for few-shot prompted models like GPT-3 and Codex that are accessed via API calls. Without a grammar, we noticed that these models explore a large number of invalid paths leading to high latency and API cost. We also found constrained decoding essential for source side prediction tasks like constituency and dependency parsing. They require each token in the input to be covered in the output. We found even fine-tuned language models struggle to learn these constraints leading to very low performance with unconstrained decoding.

Tables 2, 3 and 4 show the effect of constraints while generating from fine-tuned large language models. We find that constrained decoding is most beneficial in low-data regimes, giving on average 2.7% gain over unconstrained decoding. In the high-resource setting, the average gain is less than 1%, suggesting that the full data is nearly sufficient to learn the constraint system.

We find that constrained decoding under performs unconstrained decoding for some settings. We attribute this to the insufficient coverage of our grammars. Our grammars were induced from the training data, and thereby fail to cover novel components or combinations at test time. They are also constrained to copy quoted strings from the input utterance. However this is not strictly followed in some datasets. Table 6 shows the fraction of test outputs covered by our grammars. We could relax the grammar constraints to ensure full coverage; we leave such exploration to future work. In realistic situations, we expect grammars to be provided by domain developers ensuring full coverage.

## 5.3 Impact of Context

The datasets in BenchCLAMP require a model to use a variety of contexts. SMCalFlow, TreeDST and CoSQL datasets all have conversational context. Spider and CoSQL have database schema context which informs the target SQL prediction. BenchCLAMP allows us to perform a controlled investigation of the effect of context. Table 7 shows that while using the last agent and user utterance is helpful for all settings, the low-data regime does even better when using only the last agent utterance; without more data, training struggles to learn how to utilize (or ignore) the additional

Table 6: Grammar coverage (%) of the gold outputs in the test set of BenchCLAMP datasets.

| Dataset | Test Set Coverage % |
|---|---|
| SMCalFlow | 99.6 |
| TreeDST | 96.3 |
| MTOP (en) | 96.6 |
| Overnight (blocks) | 100.0 |
| Spider | 98.8 |
| CoSQL | 99.4 |
| AMR | 99.7 |
| PTB 3 | 100.0 |
| UD-EWT | 99.9 |

context. We find similar results for CoSQL in Table 8. Also, SQL prediction always benefits from including database values in the context along with the database schema information. We use the best settings for context for each data regime for benchmarking results in tables 2, 3 and 4.

Table 7: Lispress match accuracy of unconstrained fine-tuned T5-large with varying conversational context on SMCalFlow and TreeDST. We find more context hurts in low resource settings but helps in medium and high resource settings.

| Conv. Context | SMCalFlow | | | TreeDST | | |
|---|---|---|---|---|---|---|
| | Low | Med | High | Low | Med | High |
| No context | 37.0 | 63.8 | 72.9 | 42.4 | 68.4 | 76.0 |
| Last agent utt. | **42.6** | 70.7 | 80.6 | **59.6** | 82.7 | 87.9 |
| Last user & agent utt. | 40.0 | **71.5** | **81.3** | 58.8 | **86.2** | **90.0** |

Table 8: Test suite execution accuracy of unconstrained BART-large on CoSQL with varying context. More context hurts in low resource settings but helps in medium and high resource settings.

| Conversational Context | DB values? | Low | Med | High |
|---|---|---|---|---|
| No context | no | 21.3 | 35.9 | 34.4 |
| | yes | 25.3 | 38.9 | 40.3 |
| Last interaction | no | 24.1 | 40.4 | 39.1 |
| | yes | **28.2** | 44.2 | 44.4 |
| All interactions | no | 24.3 | 36.8 | 43.0 |
| | yes | 24.9 | **44.9** | **48.8** |

## 5.4 Few-Shot Prompting

In few-shot prompting scenario, we manipulate the context choice and ordering of examples in our prompt to Codex. The results in Table 9 show that ordering the most similar example at the end closest to the generation heads is helpful in the low-data regime, indicating that GPT-3 and Codex pay more attention to the recent past. In higher data regime, all prompt examples are almost equally relevant, hence the order does not matter as much. We find that context does not help; one of the reasons being that we can fit fewer examples in the prompt if we include context for each example.

We experiment with BM25 and similarity models for prompt retrieval for few-shot prompting. For similarity models, we pick top models from each of the three categories in SentenceTransformers leaderboard [29]: all-mpnet-base-v2, multi-qa-mpnet-base-dot-v1 and sentence-t5-xxl [25]. We find that SentenceT5-xxl surpasses other similarity models for prompt retrieval. SentenceT5 outperforms BM25 in low resource settings, but performs relatively worse when more data is available.

Table 9: (Left): Lispress match accuracy of the few-shot prompted Codex DaVinci model on SMCalFlow with different prompt order and conversational context. More context hurts few-shot prompted models. Ordering the most relevant examples closer to the generation heads improve performance. (Right): Lispress match accuracy of Codex on SMCalFlow with different prompt retrieval methods. We used `code-davinci-001` with best last prompt order and no context.

| Prompt Order | Conv. Context | Low | Med |
|---|---|---|---|
| Random | No context | 35.7 | 52.0 |
| Best First | No context | 34.3 | **53.0** |
| Best Last | No context | **36.7** | 52.0 |
| Best Last | Last agent utt. | 34.0 | 41.0 |
| Best Last | Last user & agent utt. | 26.0 | 31.0 |

| Prompt Retrieval Method | Low | Med |
|---|---|---|
| BM25 | 36.7 | **55.0** |
| all-mpnet-base-v2 | 38.0 | 49.0 |
| multi-qa-mpnet-base-dot-v1 | 38.3 | 50.0 |
| sentence-t5-xxl | **39.3** | 52.0 |

Table 10: Effect of different grammar induction data on the Lispress match constrained decoding accuracy of fine-tuned T5-large.

| Constraint Grammar | SMCalFlow | | TreeDST | |
|---|---|---|---|---|
| | Low | Med | Low | Med |
| Unconstrained | 42.6 | 71.5 | 59.6 | 86.2 |
| Induced from train split | 45.6 | 73.1 | 62.3 | 87.2 |
| Induced from full train | 46.3 | 73.1 | 64.2 | 87.2 |

## 5.5 Grammars induced from Less Data

The grammars for SMCalFlow, TreeDST and MTOP were induced using the full train dataset. This grammar is then used even with low and medium resource train splits. We expect the grammar will be provided by a developer of the domain and hence will cover all valid representations. For completeness, we report here the impact of using grammar induced from the corresponding train sets. Table 10 shows the results of decoding with train split induced grammar, and compares the performance with unconstrained decoding and decoding with grammar induced from full train set. The gains from constraints drop by $1 - 2\%$ for low resource splits when using train split induced grammar instead of full train induced grammar. It does not affect results for medium resource splits.

## 6 Conclusion

We introduce a benchmark comprising nine parsing datasets with varying target representations. We support few-shot prompting, fine-tuning and constrained decoding for all autoregressive language models and sequence-to-sequence models on these datasets. We hope that this work will encourage language model developers to consider parsing tasks as a test-bed in future work.

## 7 Limitations

Our benchmark includes data in multiple languages (all languages included in the MTOP dataset) but we only evaluate on English datasets due to compute constraints. Few-shot prompted experiments were evaluated on relatively small test sets on three datasets due to API cost limitations. As a result, we noticed high variance in the results (see Appendix section A for variance results).

## Acknowledgements

Thanks to Yunmo Chen who helped set up Llama-2 models for our benchmark experiments.

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

# A Variance of Results

All low-resource results are a mean of three training data splits. Table 11 reports the average standard deviation for each model over the three low resource splits. We find a high standard deviation of GPT-3 and Codex; one of the factors being the small size of the test set (100 examples). Fine-tuned models show relatively low variance, consistently having standard deviation lower than 2%.

Table 11: Standard deviation of the scores for each language model over the three low resource splits.

| Model | Avg. Standard Deviation |
|---|---|
| GPT-3 | 4.7 |
| Codex | 3.2 |
| T5-base | 1.2 |
| CodeT5-base | 1.1 |
| BART-large | 1.4 |
| T5-large | 2.0 |
| T5-3B | 1.5 |

# B Compute Details

We used Microsoft Azure to run all our experiments. For experiments with T5-base, CodeT5-base and BART-large, we used a single V100 GPU with 32 GB memory. We used 2 V100 GPUs for fine-tuning T5-large and 4 GPUs for fine-tuning T5-3B. Training time ranged from 2 to 6 hours depending on the size of the model.

# C Dataset and Model License and Privacy Implications

The licenses of the datasets included in BenchCLAMP are listed in Table C. We could not find a license for MTOP, but the dataset was made freely available by the creators. Our benchmark is available under the MIT licence at `https://github.com/microsoft/semantic_parsing_with_constrained_lm`.

The datasets in BenchCLAMP either use sentences from news articles or utterances generated by crowd-workers following instructions to simulate user interactions. No real user interaction data was used to generate these datasets. As a result, there is no privacy risk in using these datasets.

For fine-tuning experiments, we use pre-trained models from Huggingface. All models were released under the Apache 2.0 license.

Table 12: Licenses of datasets covered by BenchCLAMP.

| Dataset | License |
|---|---|
| SMCalFlow [1] | CC BY-SA 4.0 |
| TreeDST [10] | CC BY-SA 3.0 |
| MTOP [18] | – |
| Overnight [45] | CC BY-SA 4.0 |
| Spider [50] | CC BY-SA 4.0 |
| CoSQL [51] | CC BY-SA 4.0 |
| AMR 2.0 [4] | LDC License |
| PTB 3 [23] (Constituency parsing) | LDC License |
| UD-EWT [52] (Dependency parsing) | CC BY-SA 4.0 International |

