# OpenReview forum: "BenchCLAMP: A Benchmark for Evaluating Language Models on Syntactic and Semantic Parsing"
_NeurIPS.cc/2023/Track/Datasets_and_Benchmarks — NeurIPS 2023 Datasets and Benchmarks Poster_

### Official Review · Reviewer_ZUCA · 2023-07-17
**paper review**

**Rating:** 6
**Confidence:** 3
**Correctness:** The claims made in the paper are corr…
**Clarity:** The paper is well written.

**Strengths:**

* The collection of tasks included in BenchCLAMP gives a comprehensive picture of the performance of a language model for constrained generation.
* The methodology for constructing the datasets and fine-tuning and evaluating is well-motivated.
* The paper shows that fine-tuned encoder-decoder language models remain the best performing models for semantic and syntactic parsing, ahead of pretrained large models like GPT-3. This suggests an interesting target for improving language models. These models also perform close to or better than state-of-the-art parsing methods on the datasets included in BenchCLAMP.


**Additional Feedback:**

No additional comments.

**Documentation:**

The datasets are well documented.

**Ethics:**

No ethical concerns.

**Limitations:**

The limitations section of the paper is fair and well-written

**Opportunities For Improvement:**

* It would be good to evaluate some of the most recent pre-trained very large language models, like GPT-4, Bard, etc. It would also be interesting to study how model scale affects performance on this benchmark
* Can performance on this benchmark be related to other interesting model behaviors? Part of the initial motivation for semantic parsing was to do NLP directly on the formal structures extracted from sentences. This approach is less prominent in today's research, as language models operate well with raw text, so it would be interesting to motivate the need for semantic and syntactic parsing benchmarks further.
* In line with the previous comment, the paper does not discuss whether there are advantages to using language models to do parsing, rather than using language models to directly reason over the raw text. This would be an interesting question to investigate in light of recent advances in language model capabilities.

**Relation To Prior Work:**

Prior work is discussed clearly.

**Summary And Contributions:**

This paper introduces a benchmark consisting of several pre-existing sematnic and syntactice parsing tasks. They evaluate a range of language models and compare them with state-of-the-art constrained generation techniques. They fine-tune encoder-decoder language models on these tasks and also evaluate black-box language models through APIs. The paper also explores different generation techniques, from few-shot prompting to constrained generation.

The paper finds that language models are competitive for parsing tasks, but that these tasks remain challenging.

---

### Official Review · Reviewer_pzWP · 2023-07-21
**BenchCLAMP: A Benchmark for Evaluating Language Models on Syntactic and Semantic Parsing**

**Rating:** 8
**Confidence:** 4

**Strengths:**

Constraining LMs, in general and in particular for parsing, is a very promising research direction. I believe that this benchmark will be useful to the NLP community.
In addition, the experiments are interesting and well presented.

**Additional Feedback:**

l.128 "allowing tokens to appear in the order seen in the utterance", I guess what is meant is "appear ONLY".
l.146, "Unless otherwise state", missing "d".

**Clarity:**

Yes, is general, but concerning the grammars (section 3.2), I think the paper would benefit more precision.
Maybe (in Appendix), for each formalism, one example and the grammar inferred from this example could be given.

Additionally, I have spotted a couple of incoherences (I think):
(i)
l.77-84; "For each dataset, we create the following splits: [… We similarly create a medium-resource train split …]"
vs
l.95-96; "The Overnight train set was already small (< 5k examples), so we do not have a separate medium split for it."
So, l.77, is it really for each dataset?

(ii)
l.86-87; "we sample 2000 examples from the test sets of SMCalFlow"
vs
l.92-93; "For datasets that do not release public test sets, like SMCalFlow"
What are the test sets mentioned in l.86-87?

**Correctness:**

I'm not a big fan of how the linearisation of the dependency parses is done. Because no index is used to identify the tokens, the linearisation process is lossy.
It may be that LLM are not very good at dealing with indices, but this is their problem, not a problem of the data!
Maybe two versions of the parses could be released? (The one described and a lossless version.)

**Documentation:**

(everything is fine)

**Ethics:**

(everything is fine)

**Limitations:**

(everything is fine)

**Opportunities For Improvement:**

(See Correctness for a minor concern.)

**Relation To Prior Work:**

(everything is fine)

**Summary And Contributions:**

The authors provide the tools and data to benchmark constrained LM syntactic and semantic parsers on nine datasets, together covering seven formalisms.
They evaluate a variety of LMs and study the impact of model size, quantity of training data, CFG-based decoding constraints and prompt design.

---

> ### Author Response · Authors · 2023-08-16
>
> Thanks for your review. We address specific questions raised here:
>
> 1. Lossy dependency parse linearization: We will release a loss less linearization and a grammar supporting the representation with our camera ready version.
>
> 2. Clarity: We appreciate feedback on the grammar creation section. We will clarify the grammar creation process and add examples as suggested.

---

### Official Review · Reviewer_1bZU · 2023-07-22
**The paper proposes a benchmark for evaluating the syntactic and semantic parsing abilities of language models**

**Rating:** 6
**Confidence:** 3
**Correctness:** Yes
**Clarity:** Yes

**Strengths:**

1. This paper is well-written and easy to understand. For example, the preprocessing for each dataset is described clearly.

2. This paper conducts a comprehensive model comparison, including five encoder-decoder models that require fine-tuning and two prompt-based models. In addition, it is shown that there is a significant improvement when grammar constraints are added to the models.

**Additional Feedback:**

What are the advantages of creating this benchmark compared to using existing datasets directly? Why is it necessary to conduct parsing tests on so many datasets?



**Documentation:**

Yes

**Opportunities For Improvement:**

1. This paper does not explain well why it is necessary to evaluate the syntactic and semantic parsing abilities of LLMs. It is clear that current mainstream LLM models do not require any syntactic and semantic parsing features as inputs.

2. This paper only considers parsing data in English.

3. Apart from GPT-3 and Codex, more powerful models such as ChatGPT and GPT-4 are not considered.

**Relation To Prior Work:**

Yes

**Summary And Contributions:**

This paper proposes a benchmark for evaluating the syntactic and semantic parsing abilities of language models. The benchmark collects seven existing semantic parsing datasets and two syntactic parsing datasets, which can be used to assess the performance of prompt-based learning and fine-tuning language models. The paper tests seven language models, including GPT-3 and Codex.

---

> ### Author Response · Authors · 2023-08-16
>
> Thanks for your review. We address specific questions raised here:
>
> 1. Parsing data only in English: We provide data and grammars for all languages of MTOP. However due to compute constraints, we restrict to experiments in English.
>
> 2. What are the advantages of creating this benchmark compared to using existing datasets directly?: The benchmark ensures that the results are comparable for parsing tasks. Small changes in data preprocessing, grammar or decoding algorithm might otherwise affect comparison of language models.
>
> 3. Why is it necessary to test on so many datasets?: The output meaning representations of parsing datasets can be quite distinct from each other with varying levels of complexity. We include a large variety of datasets to support accurate language model comparisons on parsing tasks.

---

### Official Review · Reviewer_T3oy · 2023-07-23

**Rating:** 6
**Confidence:** 4
**Clarity:** Yes, the paper is well written.

**Strengths:**

1. Similar to UnifiedSKG, this paper unifies several syntactic and semantic parsing benchmarks as a seq2seq task, so that LMs can be evaluated on such tasks directly. Such unification is desirable since current LMs behave like general problem solvers that output sequences. BenchCLAMP provides helpful resources for evaluating LMs on parsing tasks.
2. The authors release context-free grammars for the proposed benchmark, which sets up a standard way to use constrained decoding during evaluation.
3. BenchCLAMP features different eval settings with different number of training samples.


**Additional Feedback:**

NA

**Correctness:**

Yes, the claims are correct, and the evaluation methods and experiment design are appropriate

**Documentation:**

Yes

**Limitations:**

Yes, the authors have discussed limitations in Section 7.


**Opportunities For Improvement:**

1. The authors only evaluate GPT-3 series on the benchmark, while I think GPT-3.5 model series should be included as well (e.g. text-davinci-003, ChatGPT).
2. I feel the models evaluated in this paper are kinda out of date. To make important  baselines for the proposed benchmark, I think llama should be also included in the training setting to give better baselines for future users.


**Relation To Prior Work:**

Yes

**Summary And Contributions:**

This paper constructs a benchmark to evaluate language models on syntactic and semantic parsing. BenchCLAMP does not collect new datasets, but ensemble nine existing datasets to form the test data. The authors linearize the output structure in a unified way so that LMs can produce them. Also, the paper uses different splits to construct several evaluation settings of distinct resources in terms of the number of training examples. To facilitate constrained decoding, the authors release context-free grammars for all datasets. Evaluation of several LMs and some empirical analysis are presented.

---

### Author Response · Authors · 2023-08-16
**Addressing Common Points Raised**

We thank the reviewers for their appreciation of our work and helpful suggestions for improvement. We address the common points raised here; we also separately answer specific questions as comments to reviews.

1. Evaluation of latest models like ChatGPT, GPT-4, Llama

We aimed to select a wide variety of language models to show the usefulness of our benchmark in various settings and to provide some trends in performance. We explored evaluating ChatGPT and GPT4, but the OpenAI API available to us for these models currently do not provide next token distribution or probabilities for the sampled tokens. As a result, it is not possible to perform beam search with constrained decoding for these models. We will provide results for the Llama model in the camera ready version.

2. Why is semantic parsing evaluation required

We believe that semantic parsing is still very relevant today even with the rise of LLMs. Although it is not used to provide input features, it is still the best way to connect language to actions or sequence of API calls. For example, the CalFlow and TreeDST datasets map input language to API calls that the system needs to make to follow the user instruction. Similarly, a system for instruction following for a robot will typically map the input instruction to a semantic parse representing the sequence of actions that the robot should execute.

---

### Decision · Program_Chairs · 2023-09-22

**Decision:**

Accept (Poster)

**Comment:**

This paper assembles a set of 7 semantic parsing and 2 syntactic parsing datasets, together with CFGs for parsing their meaning representations, and constraint decoding code to generate valid outputs from these grammars. The linearize the meaning representations for use with autoregressive sequence models. They provide evaluation code for LMs that are fine-tuned or are prompted, and provide different splits for training/evaluation for the purpose of standardization. They test 7 LMs on this benchmark - including GPT-3 and Codex.

Strengths:
1. Well written, easy to read, well motivated, and good limitations section.
2. Comprehensive i.e. Includes both seq2seq and decoder-only models, and includes models that are fine-tuned, as well as models that are prompted.
3. Has a strong takeaway - i.e. fine-tuned seq2seq still outperform pretrained models + prompting on semantic/syntactic parsing tasks.

Weaknesses:
1. Newer models such as Llama, ChatGPT, GPT-4 are not included. The authors have promised to include Llama in their camera-ready, and are unable to present results from the OpenAI models since the returned output is insufficient to perform constrained decoding. This is a good resolution of this concern by the authors.
2. An analysis of how performance changes with model scale would improve the impact of this paper.
3. Models can perform end-tasks today without being aware of syntactic/semantic parses, so this might limit the utility of this benchmark. Nevertheless, these classic tasks remain of interest to the NLP community, so this may not be too big of a concern.